# A Neural Network Approach towards Generalized Resistive Switching Modelling

**DOI:** 10.3390/mi12091132

**Published:** 2021-09-21

**Authors:** Guilherme Carvalho, Maria Pereira, Asal Kiazadeh, Vítor Grade Tavares

**Affiliations:** 1Institute for Systems and Computer Engineering, Technology and Science (INESC TEC)—INESC Technology and Science and FEUP—Faculdade de Engenharia, Universidade do Porto, Campus da FEUP, Rua Dr. Roberto Frias 378, 4200-465 Porto, Portugal; guilherme.l.leitao@inesctec.pt; 2CENIMAT/i3N, Departamento de Ciências dos Materiais (DCM) and Center of Excellence in Microelectronics and Optoelectronics Processes of the Institute for New Technologies’ Development (CEMOP/UNINOVA), Faculdade de Ciências e Tecnologia (FCT NOVA), Universidade NOVA de Lisboa, 2829-516 Caparica, Portugal; mel.pereira@campus.fct.unl.pt (M.P.); a.kiazadeh@fct.unl.pt (A.K.)

**Keywords:** resistive switching, artificial neural network (ANN), a-IGZO, device modelling

## Abstract

Resistive switching behaviour has been demonstrated to be a common characteristic to many materials. In this regard, research teams to date have produced a plethora of different devices exhibiting diverse behaviour, but when system design is considered, finding a ‘one-model-fits-all’ solution can be quite difficult, or even impossible. However, it is in the interest of the community to achieve more general modelling tools for design that allows a quick model update as devices evolve. Laying the grounds with such a principle, this paper presents an artificial neural network learning approach to resistive switching modelling. The efficacy of the method is demonstrated firstly with two simulated devices and secondly with a $4\;{\mu m}^{2}$ amorphous IGZO device. For the amorphous IGZO device, a normalized root-mean-squared error (NRMSE) of 5.66 × 10^−3^ is achieved with a [2, 50,50 ,1] network structure, representing a good balance between model complexity and accuracy. A brief study on the number of hidden layers and neurons and its effect on network performance is also conducted with the best NRMSE reported at 4.63 × 10^−3^. The low error rate achieved in both simulated and real-world devices is a good indicator that the presented approach is flexible and can suit multiple device types.

## 1. Introduction

It has long been known that complementary metal-oxide semiconductor (CMOS) technology is set to reach a downscaling limit beyond which the technology is no longer viable. As we edge closer and closer to this theoretical limit, new nanoscale technologies are starting to be pursued as a possible complement to the already existing ones. Among such new technologies, resistive switching (RS) has been labelled as one of the most promising. Known since the 1960s [1], resistive switching is a phenomenon exhibited mostly by metal oxides that consists of a reversible change of conductance of a material by the application of an electric stimulus. These devices are mostly interchangeably named memristors or resistive RAMs (ReRAM) and have been studied by many research groups in recent years, not only due to their potential in high-density non-volatile memory applications, but also as a result of showcasing similar behaviours to biological synapses, which could contribute to the development of artificial learning systems. Despite their potential, memristors are still at their infancy and there is still much to be done before they can be widely used in consumer electronics.

Memristors are typically organized in crossbar structures. Due to the passive nature of the memristor, coupled with multiple resistive levels associated with the devices, memristor crossbars are vulnerable to sneak-path currents [2]. Sneak-path currents cause reading/writing errors to the crossbar array by introducing cross-talk between devices. A known effective and simple solution to the problem is coupling memristors with transistors (1T1R). However, memristors are generally made from materials that are not compatible with CMOS transistors, such as crystalline and amorphous metal oxides or perovskites. Several solutions have been proposed to tackle the issue of integrating both memristors and transistors. Traditional solutions consist of a hybrid approach where resistive switching devices are deposited on top of standard CMOS technology and devices are connected with one another via micro/nano wire interconnects known as a through-silicon vias (TSV). In spite of having a simpler implementation, TSV lack the required robustness and may result in failure [3]. In more recent years, monolithic 3D integration has been regarded as another promising technology [4]. Other solutions opt for patterning ReRAM cells in between metal layers [5]. Nevertheless, in spite of promising to solve the 1T1R problem, 3D integration presents a very steep step in complexity and will considerably increase fabrication costs.

Modelling memristor behaviour with high enough precision, so that they can be simulated for chip design, is also an object of major concern. Current modelling approaches of ReRAMs can range from simple dynamical equations that can be easily implemented in circuit simulators (e.g., SPICE) to fairly complex systems that attempt to fully simulate ionic motion, believed at this moment to be the phenomenon responsible for resistive switching.

In general, circuit modelling of these devices has been highly influenced by the theory of Leon Chua on memristive devices [6], where resistive switching devices are dynamical systems formally represented by the following set of equations:(1)I=G(w,V,I)×V
(2)$$\frac{dw}{dt}=f\left(w,V,t\right)$$
where *w* represents an internal state of the device, *V* is the voltage applied to the terminals, and *I* is the current flowing through the device. It is worth mentioning that these equations describe a voltage-controlled device. In the case where the memristor is current-controlled, the voltage dependence should be exchanged with the current. Nowadays, some non-physical models end up making the assumption that Equation (Equation 2) can be further simplified by two separate orthogonal equations:(3)$$\frac{dw}{dt}=g\left(w,V,t\right)\times h\left(V,t\right)$$
in which h(V,t) is a function tracking how the internal state changes with electric stimuli and g(w,V,t) represents the so-called window function, responsible for binding the internal states and endowing additional non-linear behaviour.

Equation (Equation 1) is most often modelled empirically or based on conduction mechanisms particular to metal-insulator-metal (MIM) and metal-semiconductor-metal (MIS) type structures. HP labs is known as one of the pioneering groups in resistive switching, having proposed in [7] a model based on ionic drift and linear current-voltage (I-V) relation, bound by a low resistive state (LRS) and high resistive state (HRS). This work in particular triggered a set of other articles improving upon the original model, namely [8,9,10], which set to design window functions that could better describe the behaviour of real devices. In [11], Pickett concluded that the dominant conduction mechanism was due to carriers tunnelling through a barrier and thus used the Simmons tunnelling formula for MIM junctions to describe the I-V curves. Yakopcic used the same principle in his model formulation [12]. It is also well-known that some types of resistive switching are more afflicted by stochasticity. This is notably true for filamentary type memristors since conductive filament formation is non-uniform by nature and rupture might occur unpredictably [13]. Often probabilistic models have to be taken in consideration to account for device stochasticity [14].

Empirical and data-driven models have also been widely used, not only due to the complexity that physical models incur in, but also because they are sometimes more practical to develop, especially when the underlying mechanisms are unknown. With the intention of having a generalised model, Kvatinsky proposed both a current-controlled and voltage-controlled threshold-based system to describe the switching properties of memristors, known as the TEAM and VTEAM model, respectively [15,16]. Contemplating the possibility of a linear relation for the current-voltage pair, as seen in [7], Kvatinsky also proposed an exponential expression, broadening the application and providing good comparisons with some previously mentioned works. In [17], Messaris introduced a more data-driven approach that seems to achieve good results for TiO_x_-based devices.

As a matter of fact, most models found in the literature have only been tested with titanium oxide-based or hafnium oxide-based devices; however, there is a wider array of metal oxides that have been used with resistive switching properties, such as ZTO and IGZO [18,19]. These materials are of particular interest since they have been at the forefront of thin-film transistor (TFT) technology given that they not only offer one of the highest electron mobilities achievable thus far for this type of technology, but they are also key semiconductor materials in the transparent and flexible electronic market. IGZO is thus a very good candidate to achieve 2D memristor-transistor integration using the same active material. Compared to CMOS, fabrication costs would also be considerably reduced, since TFT fabrication has been widely reported to be performed at room temperatures. Another particular feature of IGZO-based devices is that they can present interface-type resistive switching instead of the more commonly found and modelled filamentary type. Unlike the latter, interface-type resistive switching provides better device-to-device reproducibility and cycle-to-cycle uniformity, which can result in higher accuracy pattern recognition applications [20]. Furthermore, non-filamentary RS is area-dependent, making it a lot more attractive for scalable circuit design.

One possible solution that has not been widely explored is capturing the internal dynamics of the memristor using an artificial neural network (ANN). ANNs are loosely inspired in brain networking and known learning principles. Backpropagation is, however, the most widely used training algorithm which has proven, in recent years, to be a very useful and powerful tool in complex system identification. Different types of neural networks exist, ranging from simple feedforward or recurrent models to more complex deep learning systems, such as GAN, deep convolutional networks, or autoencoders, to name a few [21]. Neural networks have been used successfully in the past to model other types of electronic devices, namely a-IGZO transistors [22]; however, to our knowledge, only one work so far has been published on resistive switching modelling with ANNs [23] with very little detail being provided. In this article, we explore how interface-type amorphous oxide semiconductor resistive switching can be modelled by neural networks capable of simulating dynamic systems. This data-driven model has the potential advantage of describing the behaviour of manufactured memristive devices quickly and with few measurements while offering a controlled trade-off between accuracy and computational complexity.

## 2. Materials and Methods

### 2.1. Modelling of Memristor Dynamics with Neural Networks

By analysis of Equations (Equation 1) and (Equation 2), one can see that the first equation is static and only the second is dynamical. Therefore, Equation (Equation 1) can be physically modelled if transport mechanisms are well-known or empirically approximated for LRS and HRS states. Equation (Equation 2) presents itself as more challenging. To fully model the devices’ dynamics and prevent the common pitfall of not including the memory of past dynamics in the device model [24], it is required of the network to be capable of learning data-driven dynamic systems. Since the dynamics are not known, one possible solution relies on multi-step neural networks (MSNN) [25]. In its simplest form, an MSNN consists of a multilayer perceptron structure followed by a linear multi-step method with *M* steps. Feeding the system with temporal data-snapshots will result in the learning of the underlying equation that governs the dynamics of a system, without committing to a particular class of basis functions or using temporal gradients directly. Unlike that reported in [25], using the backward differentiation formulas (BDF) with a step size of M=1 provided the best results. As such, in future references to the network, BDF with a single step-size should be assumed as the chosen multi-step method. To further explain how the system works, let us assume a simple dynamical system of the form:(4)$$\frac{\partial x}{\partial t}=k\left(x\left(t\right)\right)$$

The objective is to discover *k* without any prior knowledge of it besides the evolution of the state x(t) at time instances $t_{1},t_{2},\ldots ,t_{N}$. To that effect, a linear multi-step method can be employed, with the form:(5)$$\sum_{m=0}^{1}\left(\alpha_{m}x_{n-m}+\beta_{m}\Delta tk\left(x_{n-m}\right)\right)=0,\;n=1,\cdots ,N$$
for M=1. Here, $\alpha_{m}$ and $\beta_{m}$ are parameters specific to each multi-step scheme and $x_{n-m}$ is the state of the system *x* at time $t_{n-m}$. Finally, by placing a neural network at *k*, the dynamics of the system can be learned by minimizing the mean square error loss function:(6)$$min\left(\frac{1}{N}\sum_{n=1}^{N}{\left|y_{n}\right|}^{2}\right)$$
where
(7)$$y_{n}=\sum_{m=0}^{1}\left(\alpha_{m}x_{n-m}+\beta_{m}\Delta tk\left(x_{n-m}\right)\right)$$

One required adaptation to the network was the addition of an exogenous input to the network. This was performed since Equation (Equation 2) is not only dependent on the current state, but also on the applied voltage to the system. With the exogenous inputs, the system can then be formally described by a change in Equation (Equation 4), and subsequently Equation (Equation 7), such that for our particular case they take the form:(8)$$\frac{\partial \omega }{\partial t}=k\left(\omega \left(t\right),V\left(t\right)\right)$$
(9)$$y_{n}=\sum_{m=0}^{1}\left(\alpha_{m}\omega_{n-m}+\beta_{m}\Delta tk\left(\omega_{n-m},v_{n-m}\right)\right)$$
where $\omega_{n-m}$ and $v_{n-m}$ are the internal state of the device and the voltage applied to the terminals of the ReRAM at time $t_{n-m}$. Figure 1 showcases a diagram of the implemented MSNN.

### 2.2. Fabricated Devices

The fabricated memristive device to be modelled consists of a 60 nm IGZO film which is stacked between molybdenum and gold. At the top contact interface, a thin layer of titanium is deposited firstly for better adhesion of the gold contact. Furthermore, the titanium is an oxygen-getter layer, and as such, it reacts with the oxygen of the layer below, generating more oxygen vacancies at the top interface and providing the ohmic side of the resistive switching device. Thus, the thickness of the intrinsic IGZO should be considered lower than 60 nm. On the other hand, the junction between IGZO and molybdenum forms a Schottky barrier, granting a slight rectification on the current-voltage characteristic. The layering structure of the device is presented in Figure 2a. Devices were fabricated in a cross-point structure with an area of 4 ${\mu m}^{2}$ and a micrograph of a similar device to the one in use is shown in Figure 2b. The I-V characteristic of the device, visible in Figure 2c, exhibits bipolar switching and set occurs on the forward direction. Current state is also changed gradually without any sharp transitions to be observed. The smoother transitioning between states can be attributed to the area-dependent nature of IGZO resistive switching. Slight changes in the characteristic can be seen in the 100 cycle endurance study (see Figure 2c). Nonetheless, stabilization of the characteristic curve is achieved over time. The device presents only minor changes from cycle 50 to 100.

### 2.3. Data Sourcing and Measuring Instrument

A Keithley 4200 SCS was used to obtain the training and test data for this model. An ultra-fast I-V module with transient current and voltage measurements, coupled with multi-level pulsing capabilities, was used to generate data sequences. The ultra-fast module was required so that the switching dynamics could be captured with enough precision. It also provided the required flexibility to induce either potentiation or depression on the device with the required fine-grading. The measurement noise was filtered with a simple first-order auto-recursive filter.

### 2.4. Static IGZO Memristor Equation

As mentioned previously, either a physical or empirical expression has to be found to model the I-V relation corresponding to Equation (Equation 1). As the intrinsic mechanisms behind resistive switching in amorphous IGZO are still unknown, the empiric route was taken. The chosen equation is of the following form:
(10a)$$I=\alpha V\exp\left(-\frac{\beta }{\sqrt{V+\beta /\lambda }}\right)$$
(10b)$$\alpha =x_{0}\exp\left(x_{1}\beta \right)-x_{2}$$
where λ, $x_{0}$, $x_{1}$, and $x_{2}$ are parameters and β is defined as the internal state of the device. Note that Equation (Equation 200) is empirical (as explained below) and assumes an important role by allowing Equation (Equation 210) to be dependent on a single state variable (β). It was inspired in the field emission formulation by Padovani and Stratton [26], mostly because it provided a good mix between a diode-like behaviour at lower voltages and linear response at higher fields. In Figure 3, the results of fitting Equation (Equation 210) to some of the measured data points can be found as well as the exponential fitting between α and β on the inset graph. At first, Equation (Equation 210) is fitted to the acquired data using two independent variables, α and β. It should be noted that data acquisition is performed at fast speeds to prevent an internal state change of the device, meaning that each I-V curve in Figure 3 has a fixed value of state variable β (and α). Later, Equation (Equation 200) is used in the fitting of the parameters $x_{0}$, $x_{1}$, and $x_{2}$ that are obtained from α and β values found for each of the I-V fitting curves (the filled red dots represent the samples in the inset graph).

## 3. Results and Discussion

This section presents the results of the proposed modelling method. Firstly, the behavior of the model is demonstrated with two simulated devices.

Secondly, the method is validated with measurements taken from an amorphous IGZO device. The figure of merit used is the normalized root-mean-squared error (NRMSE) defined by:(11)$$NRMSE=\sqrt{\frac{\sum {\left({\hat{q}}_{n}-q_{n}\right)}^{2}}{\sum {\left(q_{n}-\bar{q}\right)}^{2}}}$$

Here, $q_{n}$ and ${\hat{q}}_{n}$ represent the output of the system and the desired value for the nth sample. $\bar{q}$ is the average value of the samples.

### 3.1. Learning of a Simulated Device

As a proof of concept, the proposed modelling method was tested with data generated from two different models, of two different devices, where the model parameters themselves were extracted from actual measured data. This means that the base models should be good representations of real devices. The VTEAM model and the Prodromakis window function were used to replicate a hafnium-based device [27] and the Yakopcic model was used to replicate a titanium-based memristor [28]. The parameters for the models were extracted from [16] and [12], respectively. The Prodromakis window function was added to the VTEAM model with parameters p=0.5 and j=1. A series of signals were applied to the simulated devices with different starting points and waveforms (DC, triangle, and pulse). The original signals, totalling 900 samples, were further subdivided into 9000 samples to improve training, and 80% of the data was used in training, and the remaining 20% to test the learned dynamics. The network topology used was two inputs (state and voltage), three hidden layers with 50 neurons each, and one output (next state). The normalized root mean square error (NRMSE) for the total of the 1800 samples used in the Pt-Hf-Ti memristor testing was 1.22 × 10^−5^. The titanium-based device presented a higher NRMSE of 9.51 × 10^−3^.

Regarding the hafnium-based device, Figure 4 illustrates the output of the network for two particular inputs, a square (Figure 4a) and a triangular wave (Figure 4b). As seen, the network was capable of learning all of the dynamics, struggling only slightly at the hard boundaries introduced by the window function and at the sub-threshold voltages contemplated by the VTEAM model (−0.53≤V≤0.5). This effect is visible in Figure 4a when the state variable approaches zero. Figure 5a,b also shows the plot of the output of the network after learning the titanium-based device. Much like in Figure 4a, the boundaries introduced by the window function are once again the major source of error. Nonetheless, the low error obtained is an encouraging telltale sign to move forward beyond the proof of concept.

### 3.2. Learning of an Amorphous IGZO Device

For the IGZO device, after mapping the current values to the corresponding internal state, an MSNN was fed with a series of measurements. The total dataset consisted in 34 measurements. After subdividing the dataset, a total of 816 time-series were used for training and 204 for testing.

Several different network structures were then put to the test, and the NRMSE results are reported in Table 1. As the table shows, for the most part, and as expected, the NRMSE reduces with both increasing network depth and width. It is clear that a shallow network, with one hidden layer, performs poorly in relation to the other configurations. This is conformal with the understanding that deeper networks tend to be more accurate. However, as complexity grows, the latter statement is valid if there is enough representative data being provided in the training phase. If the number of observations is reduced, larger networks will be more prone to overfitting phenomenons due to their higher number of parameters. This is probably the cause for the trend observable in Table 1, as the error tends to increase (or performance does not improve substantially) as the network reaches larger sizes. Note that early stopping in training is used to mitigate overfitting. The epoch resulting in the smallest error on the test set is saved as the best result. Even though it did not provide the best performance, the topology [2,50,50,1] was selected as the best alternative. This is mostly due to the trade-off between computational complexity and accuracy, a particularly important compromise when models are used in circuit-simulating tools such as SPICE.

The network demonstrated once again to be capable of learning the dynamics of the device and in providing filtering to the somewhat noisy measurements collected. Figure 6a,c displays the output of the network to triangular inputs at different frequencies and initial conditions. Figure 6b,d display the exact same data as Figure 6a and Figure 6c, respectively, as an I-V characteristic. This last form of representing the data is used because it provides a higher perception of precision at lower amplitudes. In fact, the filtering effect of the model is particularly evidenced at lower voltages. However, some deviation from the measurements is noticeable, but also expected. The smaller window size on the forward bias (see Figure 2c), alongside the cycle-to-cycle non-uniformity posed the biggest challenge to the network. Given that the collected learning data were mostly based on pulsing inputs, it is hard to judge whether the network was capable of learning the response to different types of stimuli (e.g., sine), nonetheless, the expected pinched hysteresis loop was present when the network was faced with an input similar to a DC sweep, as seen in Figure 6b. Inputs inducing potentiation/depression also produced a desired output even though no prior data had been trained at that particular pulse frequency. In Figure 7a,b, potentiation and depression data can be observed. Input pulses with a 6 ms time period at a 50% duty cycle and amplitudes of negative 3 V and positive 2 V were applied to induce either depression or potentiation, respectively. Gradual accumulation of error with time is also noticeable in Figure 7a,b. Nonetheless, such behaviour is to be expected, in both black-box and non-black-box approaches to dynamical system modelling until a bound is reached, at which point the error should reset or decrease significantly. As seen previously in Section 1, some works make use of a bounding function. For the particular learning set presented above, no window function was used either in training or testing; however, the inclusion of a bounding function is within the realm of possibilities and might even present learning improvements.

## 4. Conclusions

By having demonstrated the capability to learn a simulated hafnium-based device, a simulated titanium-based device, and a 4 ${\mu m}^{2}$ IGZO memristor, it can be said that the introduced modelling approach has the potential to adapt to multiple device types and materials. Regarding the fitting error, the hafnium-based simulated device presented a lower NRMSE than the amorphous IGZO memristor. This can be attributed to the fact that the simulated device is not subject to measurement added error and cycle-to-cycle non-uniformity. Furthermore, more data samples can be generated, facilitating learning. Notwithstanding the higher NRMSE, the IGZO device modelling achieved good results with a compact network structure, reducing computational complexity significantly. Low computational complexity is paramount to the implementation of device models in SPICE simulation.

Another very important aspect of the modelling approach was the capability of the network to learn potentiation and depression. With the increased interest in using memristors in ANN applications, both potentiation and depression assume a critical role is shaping the future of neuromorphic computing.

## Figures and Tables

**Figure 1 micromachines-12-01132-f001:**
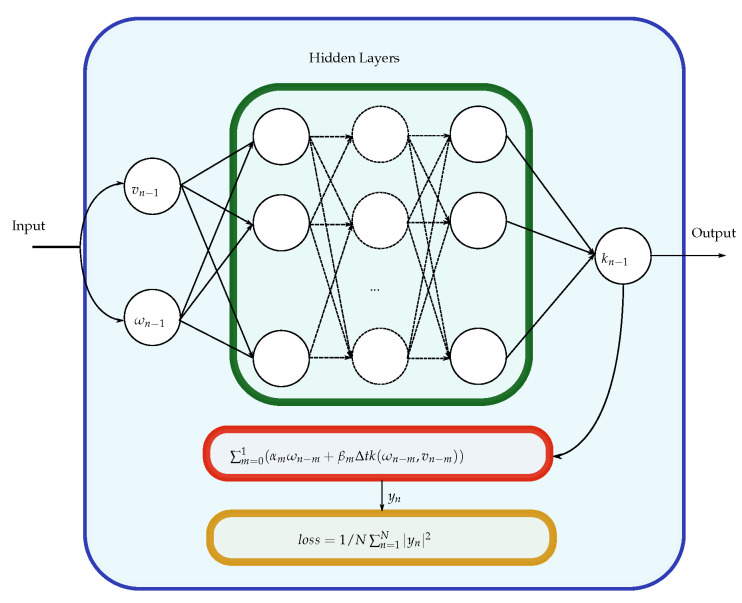
Diagram of the implemented multi-step neural networks (MSNN) with the exogenous input. During testing, the output and input of the network are fed to an ordinary differential equation solver. The output of the solver will be the new input of the network $\omega_{n}$.

**Figure 2 micromachines-12-01132-f002:**
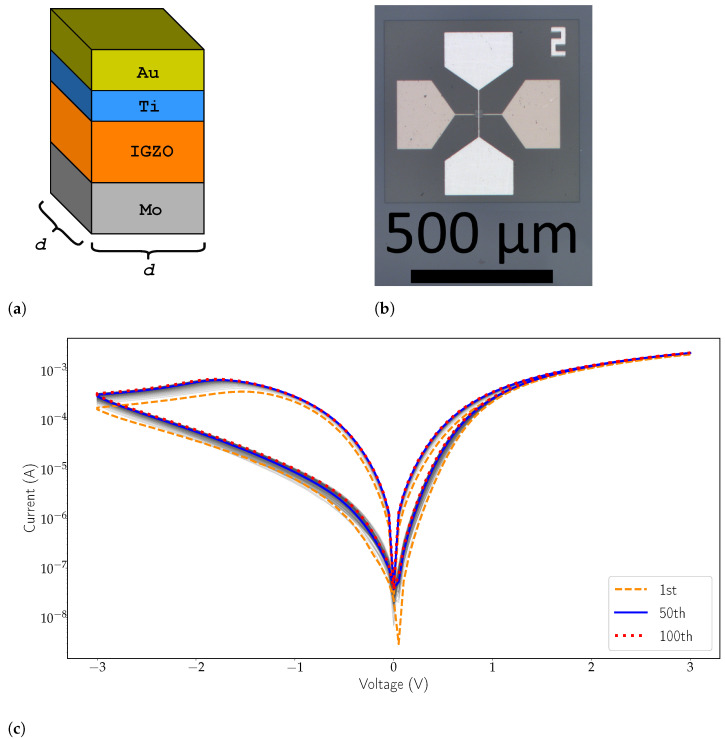
(**a**) Diagram of the IGZO memristor layers. The width of the device is represented by *d*, with the DUT having d=2μm (figure is not to scale). (**b**) Micrograph of the fabricated amorphous IGZO devices showcasing a cross-point structure with a 4 ${\mu m}^{2}$ area. The numbers represent the width of the device in μm. (**c**) Logarithmic I-V characteristic of a 4 ${\mu m}^{2}$ IGZO memristor subjected to 100 endurance cycles.

**Figure 3 micromachines-12-01132-f003:**
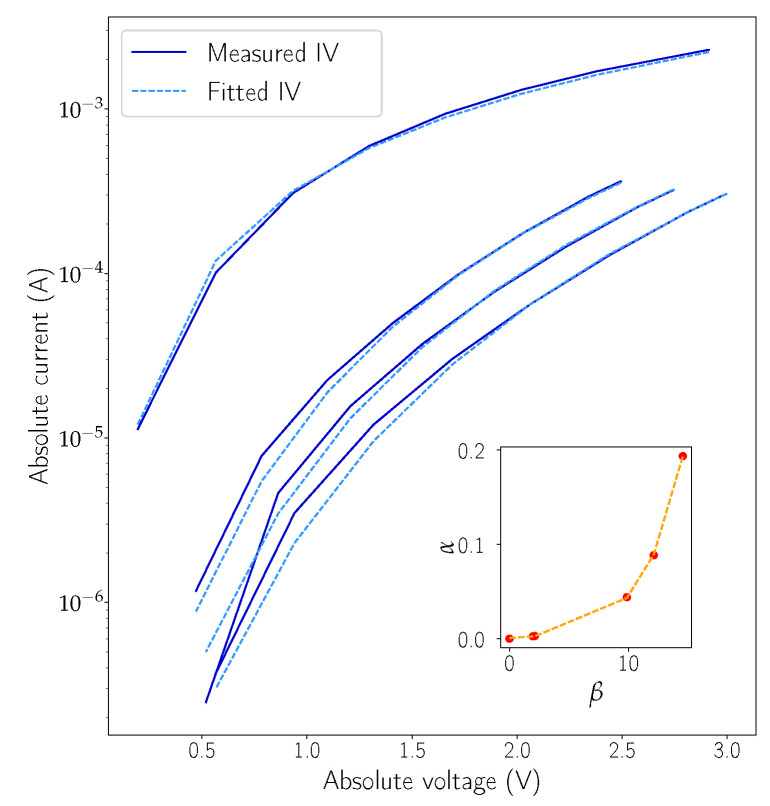
Fitting of data extracted from a 4 ${\mu m}^{2}$ IGZO memristor to the empirical Equation (Equation 210). The fitting data are used to model the static I-V characteristic of the device. The inset graph represents the α−β relation, two fitting variables related exponentially by Equation (Equation 200). The filled dots in the inset graph represent the best-fitting α and β variables for each I-V curve, and the dashed line is the result of applying the second exponential fitting defined by Equation (Equation 200). The second fitting is fundamental to prevent having two independent state variables influencing the dynamics of the device.

**Figure 4 micromachines-12-01132-f004:**
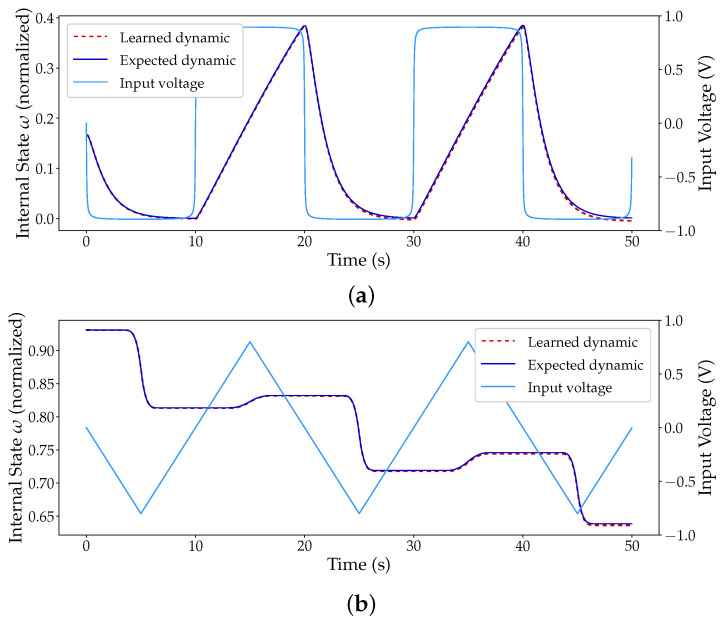
Expected dynamics of a simulated Pt-Hf-Ti memristor and the output of an MSNN network with external input and [2,50,50,50,1] topology. (**a**) Output of the network to a square-shaped periodic input. (**b**) Output of the network to a triangular-shaped periodic input.

**Figure 5 micromachines-12-01132-f005:**
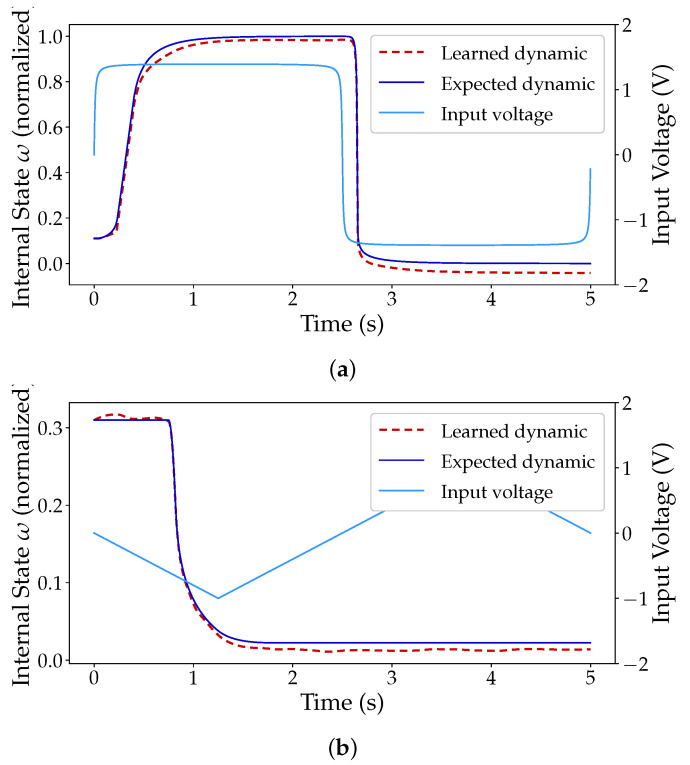
Expected dynamics of a simulated Pt-TiO_2−x_ memristor and the output of an MSNN network with external input and [2,50,50,50,1] topology. (**a**) Output of the network to a square-shaped periodic input. (**b**) Output of the network to a triangular-shaped periodic input.

**Figure 6 micromachines-12-01132-f006:**
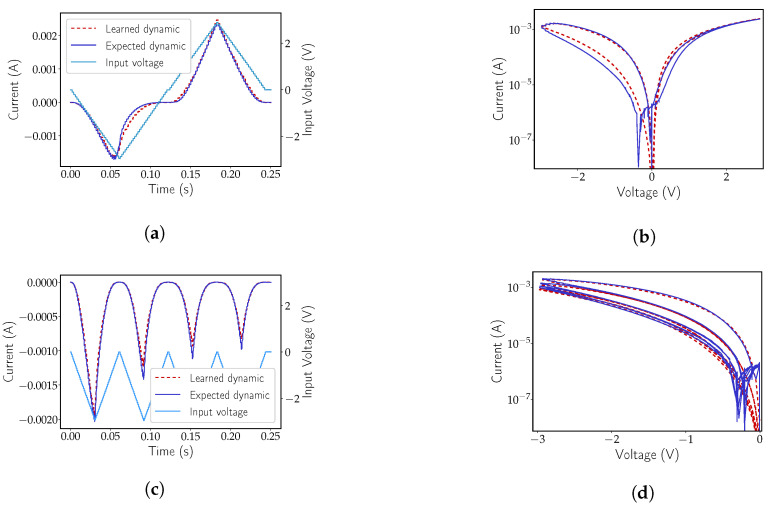
Expected dynamics of an a-IGZO memristor and the output of an MSNN network with external input and [2, 50, 50, 1] topology. (**a**,**b**) represent the same response to a single DC sweep style input, however (**a**) is represented over time and on a linear scale and (**b**) displays the logarithmic I-V plot, commonly used to show a memristor hysteresis loop. Plots (**c**,**d**) represent the output of the network to a depression inducing triangular waveform. Just like in (**a**,**b**), the left-side image, (**c**), is a linear-time plot and the right-side image, (**d**), a logarithmic I-V representation.

**Figure 7 micromachines-12-01132-f007:**
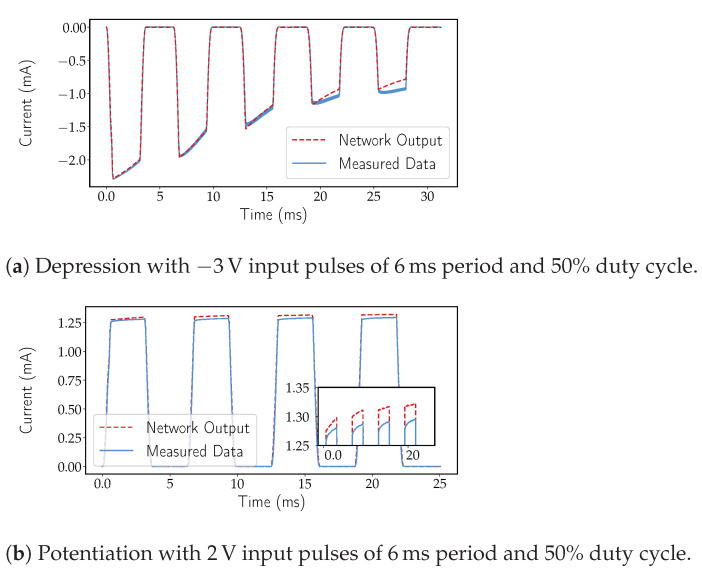
Expected dynamics of a a-IGZO memristor and the output of an MSNN network with externa input and [2, 50, 50, 1] topology to both potentiation and depression inducing pulses. (**a**) Demonstration of depression with the output of the network versus a point-cloud of measured data. (**b**) Demonstration of potentiation with the output of the network versus a point-cloud of measured data. The inset graph showcases a zoomed in portion of image (**b**), pointing out the difference between network output and measured data.

**Table 1 micromachines-12-01132-t001:** Normalized root-mean-squared error (NRMSE) values of test data for different numbers of hidden layers and neurons per layer.

Hidden Layers	Neurons per Layer			
	**15**	**30**	**50**	**100**
1	-	52.5 × 10^−3^	12.1 × 10^−3^	27.4 × 10^−3^
2	9.18 × 10^−3^	7.46 × 10^−3^	5.66 × 10^−3^	5.00 × 10^−3^
3	19.6 × 10^−3^	5.36 × 10^−3^	4.63 × 10^−3^	7.21 × 10^−3^

## Data Availability

Not applicable.

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
