# Peer review of "A Neural Network Approach towards Generalized Resistive Switching Modelling"

_micromachines, 2021, doi:10.3390/mi12091132_

Round 1

Reviewer 1 Report

In this article, the authors present an artificial neural network learning approach to resistive switching modeling. Moreover, the authors fabricated and simulated IGZO-based RRAM devices and the model is tested with the measured results. The study on resistive switching modeling is interesting. However, I cannot recommend this article in its current form. Several concerns are as follows:

  1. The abstract contains a larger part of introduction, and the real results of this study are missing. The abstract should clearly show the outputs of the study.
  2. There is no subsection 1.2. Then it isn't very meaningful to make a subsection 1.1.
  3. On page 2 line 60, it is mentioned that most of the models in the literature have been tested with HfO2 and TiO2. The authors tested this model with an IGZO-based device. What is the benefit of IGZO compared with HfO2 or TiO2? If this model was tested with HfO2 or TiO2, how the outputs of the model would have been affected?
  4. On page 5 line 125, the authors mentioned that the devices exhibited bipolar switching. However, the typical I-V characteristics of the device are missing. The typical I-V characteristics of the fabricated devices repeated for several cycles must be added to demonstrate the reliability and repeatability of the bipolar switching.
  5. In Fig 3, what information is depicted from the measured I-V and their fitting? What does the inset figure show? Proper discussion should be added.
  6. In Fig 4 and Fig 5, subfigure numbers must be added (like 4a, 4b), and each subfigure must be explained in the text.
  7. The discussion in Fig 5 is completely missing. Subfigure numbers should be added, and detailed descriptions should be added about these results.
  8. What are the different I-V cycles showing in Fig 5? What are these right-side I-V graphs showing? One graph shows few negative sweep cycles while the other shows one positive and one negative sweep cycle.
  9. Although the thickness of IGZO is quite bigger (60 nm), why is the On/Off ratio very small?
  10. What were the pulse parameters used for the measurement of depression characteristics in Fig 5b?
  11. Why were the potentiation characteristics not measured and simulated?
  12. On page 4 line 127, the authors mentioned that various cross-point devices were fabricated with different device sizes. However, the measured and simulated results of only one device are shown in this study. What was the purpose of other devices?

Reviewer 2 Report

Review

Carvalho and coauthors present a study on the application of an artificial neural network for the modulation within resistive switching. The authors did a great job in explaining the importance of their approach for future nanoscale technological switching devices. Overall the scientific discussion is well executed. However, some minor details could be adjusted to increase the comprehensibility. In addition, a problem with the presentation in some Figures should be corrected. All in all, the paper fits very well into the scope of the journal Micromachines and I recommend acceptance after major revisions, as noted below. 

Comments:

1) The introduction is brief, maybe a bit too brief. I assume that a large part of the readership is not familiar with the current challenges of the field, and therefore recommend that these be clearly stated and referenced. 

2) In Figure 4 and 5, the dashed lines of the learned dynamics is hard to make out against the expected voltage in most cases.

3) The capabilities of the artifical neural network to respond to the dynamics depends on its „brain capabilities“, i.e. the number of neurons available. The network topology was realised using 50 neurone in 3 hidden layers. It is, however, far from understood how many neurone and hidden layers are required in general. I recommend to study and present the fitness of the model using different combinations of neurons and hidden layers. Could the same performance be expected for far fewer neurons (e.g., 10, 20, 30, 40, 50) and/or hidden layers (1,2,3)? I am convinced that a clear answer to this questions would significantly improve the scientific value of this contribution. (As in biological brains, it is not the size of the brain that matters but its internal connectivity formed by training.)

Round 2

Reviewer 1 Report

The authors tried to address the questions raised by the reviewer. The current manuscript is detailed and complete so the reviewer suggests this for publication.

Author Response

We would like to thank again to reviewer #1 for helping with insightful comments that contributed to an improved manuscript

Reviewer 2 Report

Carvalho and co-authors have provided a revised version of their contribution and I am almost entirely satisfied with the changes. In particular, I welcome the addition of more information on the importance of the number of neuron and hidden layers in the network topology. Also, the change to  normalised MSE values seems reasonable. However, it seems that the discussion on the importance of the neuron number and hidden layer depth is too brief. I recommend to extend the discussion and to explain the reasons for the differences in NRMSE results. Apart from this, I recommend acceptance after this minor revision.
